# Preparation and Characterization of Regenerated Cellulose Membrane Blended with ZrO_2_ Nanoparticles

**DOI:** 10.3390/membranes12010042

**Published:** 2021-12-28

**Authors:** Xin Huang, Feng Tian, Guohong Chen, Fanan Wang, Rengui Weng, Beidou Xi

**Affiliations:** 1College of Ecological Environment and Urban Construction, Fujian University of Technology, Fuzhou 350118, China; huangx_fal@163.com (X.H.); Windt8023@163.com (F.T.); 13052696442@163.com (G.C.); fanan_wang@foxmail.com (F.W.); 2Fujian Eco-Materials Engineering Research Center, Fujian University of Technology, Fuzhou 350118, China

**Keywords:** ultrafiltration membrane, regenerated cellulose membrane, ZrO_2_, anti-fouling property

## Abstract

It is of great significance to search for efficient, renewable, biodegradable and economical membrane materials. Herein, we developed an organic-inorganic hybrid regenerated cellulose membrane (ZrO_2_/BCM) with excellent hydrophilic and anti-fouling properties. The membrane was prepared by introducing ZrO_2_ particles into an N-Methylmorpholine-N-oxide(NMMO)/bamboo cellulose(BC) solution system by the phase inversion method. The physi-chemical structure of the membranes were characterized based on thermal gravimetric analysis (TGA), Fourier transform infrared spectroscopy (ATR-FTIR), field emission scanning electron microscopy (FE-SEM), and X-ray diffraction (XRD). The modified regenerated cellulose membrane has the excellent rejection of bovine serum albumin (BSA) and anti-fouling performance. The membrane flux of ZrO_2_/BCM is 321.49 (L/m^2^·h), and the rejection rate of BSA is 91.2%. Moreover, the membrane flux recovery rate after cleaning with deionized water was 90.6%. This new type of separation membrane prepared with green materials holds broad application potential in water purification and wastewater treatment.

## 1. Introduction

With the continuous development of society, the problem of water pollution has attracted more and more attention [1]. Ultrafiltration technology plays a crucial role in water purification and wastewater treatment [2,3]. Ultrafiltration membranes materials mainly include polyethersulfone (PES) [4,5,6], polysulfone (PSF) [7,8] and polyvinylidene fluoride (PVDF) [9,10,11]. However, in the process of polymer membrane synthesis, non-biodegradable organic materials will cause environmental pollution and energy waste, so it is of great research significance to seek environmentally friendly and economical membrane raw materials [12].

Cellulose is one of the most abundant renewable and biodegradable organic materials [13,14]. Regenerated cellulose (poly(1,4)-d-glucose), obtained by dissolving cellulose, has the advantages of good chemical stability, high hydrophilicity and biodegradability, and has gradually become a research hotspot for membrane materials [15,16]. However, cellulose membranes suffer from low mechanical strength and poor anti-fouling property [17,18].

Membrane fouling is an crucial issue in ultrafiltration(UF) separation applications [19,20]. Contaminants such as proteins are easily adsorbed and deposited on the membrane surface and in the pores, which leads to a decrease in permeation flux and shortens the service life of the membrane [21]. In addition, the membrane structure with low mechanic strength would be damaged during the long-term operation of the membrane, resulting in reduced flux and poor flux recovery after cleaning. Therefore, improving the anti-fouling performance and stable performance of the membrane is still a challenge for membrane applications [22,23].

In recent years, the introduction of nano inorganic oxides, such as SiO_2_, Al_2_O_3_, TiO_2_, and ZrO_2_ into polymer membranes for improving the antifouling performance of the membrane has become one of the research hotspots [24,25,26]. The blending of these fillers would modulate the surface properties of the membrane and improves the antifouling. ZrO_2_ enjoys the merits of pleasurable thermal stability, corrosion resistance and good biocompatibility, which can improve the physical and chemical properties of the membrane [27,28,29]. Meanwhile, when ZrO_2_ is fixed on the surface of the membrane, the polarity of the Zr-O bond is easy to form a hydroxyl group (-OH) with the hydrolysis of the particle surface to improve the hydrophilicity of the membrane. Pang et al. [30] blended ZrO_2_ with PES to prepare an anti-fouling composite ultrafiltration membrane which has a higher porosity and a larger average pore size. Shen et al. [31] used a two-step method to introduce functionalized ZrO_2_ into the PVDF ultrafiltration membrane to improve the separation efficiency of the membrane for oil-water mixtures. Wen et al. [32] prepared a dense ZrO_2_ ultrafiltration membrane with nanocrystals as the precursor, which effectively reduced the pollution of ceramic membranes. It can be seen that the introduction of nano-ZrO_2_ into the membrane material improves the permeability and antifouling performance of the membrane.

However, the modification of cellulose membrane by nano-ZrO_2_ has been rarely reported. In this work, the ZrO_2_/BCM was prepared by blending nano-ZrO_2_ with natural bamboo cellulose (BC) using a convenient phase inversion method. This study focuses on the systematic analysis of the effect of ZrO_2_ on the surface microstructure, composition, crystal structure, and thermal stability of regenerated cellulose. In addition, the protein retention performance of ZrO_2_/cellulose membrane and the stability during the filtration process were also studied.

## 2. Materials and Methods

### 2.1. Materials

Cellulose with a polymerization degree of 650 was kindly provided by Sichuan Tianzhu Bamboo Resources Development Co., Ltd. (Chengdu, China). N-Methylmorpholine- N-oxide (NMMO) (Analytical reagent > 97%) was obtained from Tianjin Hainachuan Science and Technology Development Co., Ltd. (Tianjin, China). Gallic acid (PG, Sinopharm Group Chemical Reagent Co., Ltd., Shanghai, China). Zirconium Dioxide (ZrO_2_) was obtained from Macklin Co., Ltd. (Shanghai, China). Bovine serum albumin (BSA) was purchased from Aladdin Chemical Reagent Co., Ltd., Shanghai, China. The water used in this work was deionized.

### 2.2. Preparation of the Membrane 

The preparation of the membrane was schemed in Figure 1. Firstly, an appropriate amount of BC was dried at 60 °C under vacuum for 12–24 h before use. Secondly, the nano-ZrO_2_ particles (0.5–2.5 wt.%, based on BC) were added into the 85 wt.% NMMO aqueous solution and ultrasonically dispersed for 30 min [33,34]. The above mixture was then heated to 90 °C, followed by the addition of 2~3 wt.‰ of antioxidant (n-propyl gallate). The dried BC (3 wt.%, based on NMMO) was further dissolved in the mixture with continuously stirring for 2~3 h at a raised temperature of 110 °C. The resulting mixture was defoamed for 4~6 h at 90 °C to obtain a uniform brown-yellow casting membrane liquid. 

The casting membrane liquid was poured onto the non-woven fabric on the coating machine. The scraper was heated to 60~90 °C and the scraping speed was controlled to 20 cm/min. The as-scraped membrane was placed in air for 10~15 s and then soaked in deionized water for another 24~48 h [35]. Finally, the membrane was taken out and dried in the air to obtain a regenerated cellulose membrane with a thickness of 300 μm. The obtained modified cellulose regenerated membrane was recorded as ZrO_2_/BCM. The unmodified regenerated cellulose membrane was prepared as the same procedure except for the addition of ZrO_2_, and recorded as BCM.

### 2.3. Membrane Characterization

Cellulose regenerated membranes were observed on a scanning electron microscope (SEM, Zeiss Sigma 300, Carl Zeiss AG, Jena, Germany). The crystallinity of the BCM and ZrO_2_/BCM were determined by an X-ray diffraction system (Ultima IV, Rigaku, Tokyo, Japan). The test angle was set from 5° to 90° at the speed of 2.4 °/min. Infrared spectra in the 4000~600 cm^−1^ range were recorded with an FT-IR instrument (Bruker VERTEX 70 & ALPHA, ) at room temperature. The mass change of the membrane material as a function of temperature was recorded with a TG-DTA Instruments (TG209F3, NETZSCH, Selb, Germany) at a nitrogen flow rate of 20 mL/min. Approximately 2–3 mg of sample was weighed and heated from 25 to 900 °C. 

### 2.4. Performance of Regenerated Cellulose Membranes

(1)Pure Water Flux of the Membrane.

The permeation performance of the membranes was tested as shown in Figure 2. The membrane was pretreated in the system under testing conditions for 30 min at a pressure of 0.1 MPa until the pressure and water output were stable. After that, the volume of water passing through the membrane was recorded every three minutes. The water flux was calculated by Equation (1):(1)$$Q_{w}=\frac{V}{A\cdot t}$$
where $Q_{w}$ is the permeate flux (L/m^2^∙h), *V* is the volume of permeate (L), *A* is the effective area of the membrane (m^2^), and *t* is the permeate collection time (h). 

(2) Separation performance of membrane.

Ultrafiltration of membranes was carried out with bovine serum albumin (BSA) as contaminant with an initial concentration of 1000 mg/L. The concentration of BSA solution in the filtrate was measured at the 278.5 nm wavelength by a UV spectrophotometer. The rejection rate of the membrane was defined by Equation (2):(2)$$R=\left(1-\frac{C_{2}}{C_{1}}\right)\times 100\%$$
where *R* is the membrane rejection rate (%), $C_{1}$ is initial concentration (mg/L), $C_{2}$ is filtrate concentration (mg/L).

(3) Acid and alkali resistance of regenerated cellulose membrane.

The solutions with pH of 2, 4, 8, and 10 were prepared by dilution of 1 mol/L HCl or 1 mol/L NaOH. The membrane was immersed in acid and alkali solution for five days, and the change of membrane water flux was measured to analyze the acid and alkali resistance of the regenerated cellulose membrane. 

(4) Anti-fouling of regenerated cellulose membrane. 

The used regenerated cellulose membrane was placed in the membrane filtration system (Figure 2) and treated with deionized water, 0.01 mol/L HCl, or 0.01 mol/L NaOH for 0.5 h, respectively. Then, the membrane was taken out and cleaned with deionized water on both sides several times. The membrane flux recovery rate (r) was obtained by comparing the water flux before and after the cleaning procedure, which was used to evaluate the anti-fouling performance of the regenerated cellulose membrane. Where *r* is the membrane flux recovery rate (%), $J_{1}$ is the initial water flux of the membrane (L/m^2^∙h), $J_{2}$ is the water flux after membrane cleaning (L/m^2^∙h).
(3)$$r=\frac{J_{2}}{J_{1}}\times 100\%$$

### 2.5. Characterization of Porosity and Average Pore Size 

(1) The porosity of regenerated cellulose membrane was calculated by Equation (3):(4)$$\varepsilon =\frac{w_{1}-w_{2}}{\rho_{w}Sd}\times 100\%$$
where ε is membrane porosity (%), $w_{1}$ is wet membrane weight (g), $w_{2}$ is dry membrane weight (g), $\rho_{w}$ is the density of pure water (0.998 g/cm^3^) at 25 °C, S- is membrane area (cm^2^), d is membrane thickness (cm).

(2) The average pore size was calculated by Caout-erfurt Ferry equation [36], Equation (4): (5)$$r_{m}=\sqrt{\left(2.9-1.75\varepsilon \right)\times \frac{8\eta dQ}{\varepsilon \cdot \Delta P}}$$
where $r_{m}$ is average membrane pore size (nm), ε is membrane porosity (%), η is deionized water viscosity at 25 °C (8.9 × 10^−4^ Pa·s), d is membrane thickness (cm), Q is deionized water flux (L/m^2^∙h), ΔP is membrane pressure at operation (MPa).

## 3. Results and Discussion

### 3.1. The Influence of ZrO_2_ Content on the Physical Properties of the Membrane

Membrane filtration performance is a key index to judge the function of a membrane. In this study, the effect of the addition of nano-ZrO_2_ particles on the filtration performance of the regenerated cellulose membrane was investigated under a test pressure of 0.1 MPa. Ultrafiltration of the membranes was carried out with bovine serum albumin (BSA) as contamination.

The influence of ZrO_2_ dosage on the properties of regenerated cellulose membrane is shown in Figure 3. It can be observed that the addition of ZrO_2_ particles increases the porosity of the regenerated cellulose membrane. The greater the porosity ensures a high membrane flux. The mass fraction of ZrO_2_ particles increases to 1 wt.%, the water flux of the regenerated cellulose membrane continuously increases and reaches the maximum value. Its water flux is 321.49 (L/m^2^∙h), indicating that the addition of ZrO_2_ particles improved the hydrophilicity of the regenerated cellulose membrane to a certain extent. However, with the excessive content of ZrO_2_ in the casting membrane liquid system, this resulted in a decrease in the pore size of the membrane, which will lead to a sharp drop in the water flux of the cellulose hybrid membrane [32,37].

The addition of excessive nanoparticles leads to the reduction of membrane pore size and porosity. Similar experimental results were obtained previously by Shen and coworkers for the PVDF/ZrO_2_-g-PACMO hybrid membrane fabricated with the phase inversion method [31]. The decrease in pore size makes the membrane compact, resulting in a decreasing trend in the average pore size of the regenerated cellulose membrane. As shown in Figure 3, the decrease in pore size leads to an increase in the resistance of water to pass through the membrane, and the rejection rate of BSA continues to increase. 

### 3.2. The Influence of ZrO_2_ Content on the Antifouling Performance of Membrane

Through the rejection experiment of BSA under continuous operation time, the antifouling performance of ZrO_2_/BCM was evaluated. The fouling of the membrane could be originated from the effects of adsorption fouling, membrane pore blockage, steric hindrance, and/or concentration polarization [24]. The inorganic nanoparticles were dispersed uniformly in the polymer, so that the interaction between the nanoparticles and the membrane matrix made the membrane have a stable antifouling performance [38]. As shown in Figure 4, ZrO_2_ modifies the surface of the regenerated cellulose membrane, effectively resisting the deposition of contaminants on the surface and reducing the interaction force between the contaminants and the surface [32].

It can be seen intuitively from Figure 5 that before operation, the separation efficiency of BSA by the membrane is improved as ZrO_2_ particles are embedded into the membrane structure. It can be obtained from Table 1 that the rejection rate of BSA for the regenerated cellulose membrane with 1 wt.% of ZrO_2_ is 91.3%, which was decreased over time. When operating for 180 min, the BSA rejection rate of the BCM decreased by 40% to 45.3%. In the same period, the rejection rate of 1 wt.% ZrO_2_/BCM to BSA was 60% higher than that of BCM. ZrO_2_/BCM shows excellent antifouling performance. The ZrO_2_ ultrafiltration membrane prepared by Wen et al. also showed excellent resistance to BSA [32].

The change of membrane flux of BSA solution during continuous operation time is shown in Figure 6. At the beginning of the operation, BSA rapidly accumulates on the membrane surface and in the pores, resulting in a significant decrease in membrane flux. When the contaminants on the surface and inside of the membrane accumulate to a certain extent, the shear force generated by the cross-flow of the filtrate on the membrane surface will gradually reach equilibrium. The interaction between pollutants becomes the dominant factor in changing the membrane flux, and the resistance on the membrane surface tends to be flat. Therefore, the membrane flux changes smoothly in the later stage of the experiment, and the membrane flux change of ZrO_2_/BCM was below 4%. In summary, when the mass fraction of ZrO_2_ particles is 1 wt.%, ZrO_2_/BCM has good stability and rejection of BSA. Table 2 shows BCM and 1 wt.% ZrO_2_/BCM membrane performance parameters.

Hydrophilicity is an important attribute of the membrane, which directly affects the permeability of the membrane. As shown in Figure 7. The contact angle of the BCM is 43.9° ± 2.2°. Cellulose itself is highly hydrophilic. The contact angle of the 1 wt.% ZrO_2_/BCM is 33.6° ± 3.7°. It indicates that during the membrane formation process, nano-ZrO_2_ is distributed on the surface and throughout the bulk of the membrane, and the hydroxyl groups carried by them enrich the hydrophilicity of the membrane [30].

### 3.3. Characterization of Regenerated Cellulose Membrane

#### 3.3.1. SEM Observations

The microstructures of the regenerated cellulose membrane were observed by SEM scanning electron microscope. As shown in Figure 8a, the surface of the BCM membrane presents a large number of pore structures. Regenerated cellulose membranes are extremely hydrophilic, and will quickly undergo liquid-liquid stratification with water in the water coagulation bath, thereby forming larger pores [39]. Nanoparticles will spontaneously form agglomerates [25,40]. As shown in Figure 8c, after adding 1 wt.% of ZrO_2_ particles, the surface of the membrane became dense without agglomeration, indicating that the nanoparticles were uniformly dispersed in the regenerated cellulose membrane. However, with the addition of excessive ZrO_2_ particles (2 wt.%), agglomeration of nanoparticles appears on the surface of the membrane. It indicates a decrease in the average pore size of the membrane and an improvement in the structure of the membrane surface by adding nanoparticles [41].

It can be seen from Figure 8b that the BCM membrane structure has a clear layered and porous structure. It will reduce the stability of the membrane. In Figure 8d there are a large number of dispersed white spots in the regenerated cellulose membrane structure when 1.0 wt% ZrO_2_ particles are added. Under high magnification, it can be seen that the ZrO_2_ particles are uniformly attached to the surface of the membrane. The addition of ZrO_2_ reduces the macropores of the membrane structure and improves the membrane structure. However, the excessive amount of ZrO_2_ caused the blockage of the membrane structure.

The elemental composition of ZrO_2_/BCM was investigated by EDSs detection from two randomly selected points from ZrO_2_/BCM [42]. As shown in Figure 9, the C and O elements were detected, proving the chemical composition of cellulose. Meanwhile, the presence of Zr elements was also detected. Combined with the SEM images of the surface and cross-section of ZrO_2_/BCM, it is shown that ZrO_2_ nanoparticles have been uniformly dispersed in the casting membrane liquid and embedded in the regenerated cellulose membrane to improve the structure of the membrane [27,37].

#### 3.3.2. ATR-FTIR Analysis

Figure 10a shows the FT-IR spectra of BC, BCM, 1 wt.% ZrO_2_/BCM. Since intramolecular and intermolecular hydrogen bonds are generated during the dissolution of cellulose, -OH and -NH stretching vibration intensity peaks appear at 3378.03 cm^−1^ and 3354.17 cm^−1^. In the figure, there is C-H stretching at 2900 cm^−1^ and 2898.64 cm^−1^ [12]. In addition, there ais C=O stretching at 1630 cm^−1^ and C-O stretching vibration peaks at 1060 cm^−1^ in all three, which prove the presence of cellulose components [15,43]. The position of the ZrO_2_/BCM absorption peak is the same as that of BCM. There is no new absorption peak in the FT-IR spectrum. 

#### 3.3.3. XRD of Regenerated Cellulose Membrane

Figure 10b shows that there are strong diffraction peaks at the 2θ of 16.26°, 22.78° and 26.04°, corresponding to the three crystal planes (101), (002), and (040) of cellulose [44]. As the intramolecular and intermolecular hydrogen bonds of cellulose are opened during the dissolution process, the crystalline structure of cellulose is destroyed, and the crystallinity of cellulose gradually decreases during the process of dissolution and regeneration [45]. The XRD spectra showed strong diffraction peaks consistent with the characteristic peak positions of ZrO_2_ particles at 2θ of 24.5°, 28.3°, 31.5°, and 34.5° [37]. Observing the XRD spectrum of 1 wt.% ZrO_2_/BCM, no other new diffraction peaks were found. Combined with the absence of new characteristic peaks in FT-IR, it can be inferred that the ZrO_2_ particles and cellulose are combined in a blended form. In addition, the addition of ZrO_2_ particles gradually weakened the degree of crystallinity of the regenerated cellulose membrane [32,46]. It can be inferred that there is a certain force between the ZrO_2_ particles and the cellulose polymer, which changes the stress distribution of the regenerated cellulose membrane. This shows the compatibility of ZrO_2_ particles and good affinity with the membrane matrix.

#### 3.3.4. TGA Analysis

TG shows that there is evaporation of water on the surface of all three of them below 100 °C as shown in Figure 10c [47]. The initial decomposition temperatures of the three are 204.07 °C, 150.73 °C, and 201.86 °C, respectively. The results show that compared with the original bamboo cellulose (BC), the thermal stability of BCM was slightly lower, which might be caused by cellulose degradation in the dissolution and regeneration processes [48]. In addition, the residual amount of ZrO_2_/BCM after thermal decomposition is also higher than that of BCM, which proves the existence of ZrO_2_ in the membrane matrix. The addition of ZrO_2_ particles increased the initial decomposition temperature of the membrane, ZrO_2_/BCM has better thermal stability.

### 3.4. Acid and Alkali Resistance of Regenerated Cellulose Membrane

The acid and alkali resistance of membranes was further explored to expand the application of membranes in different conditions. As shown in Figure 11, the membrane flux of regenerated cellulose membrane under acid/base conditions is higher than the initial membrane flux. It may be due to the corrosion in a strong acid/base. After soaking for five days in pH 10 solution, the membrane flux of the regenerated cellulose membrane changed the most, in which the flux of BCM and 1wt.% ZrO_2_/BCM reached 447.13 L/m^2^ h and 412.7 L/m^2^ h, respectively. Compared with the initial membrane flux, the membrane flux of 1 wt%ZrO_2_/BCM has a smaller change compared with BCM. It indicates that the addition of ZrO_2_ particles improves the acid and alkali resistance of the regenerated cellulose membrane.

### 3.5. Anti-Fouling of Regenerated Cellulose Membrane

It is necessary to clean the contaminated membrane to improve the ultrafiltration performance of the membrane.

Figure 12 shows the membrane flux recovery rate after cleaning the used regenerated cellulose membrane under different conditions. As can be seen, the membrane flux recovery rate of the regenerated cellulose membrane basically reached 90% by washing with HCl and NaOH. The membrane flux of BCM could hardly be recovered with water as the washing agent, and the recovery was about 65%. On the other hand, the membrane flux of 1 wt.% ZrO_2_/BCM after cleaning with deionized water reaches 290.8 (L/m^2^h). It also has a recovery rate of 90.6%, 24.6% higher than that of BCM. It was rationalized that the addition of ZrO_2_ effectively reduces the contact of pollutants with the membrane surface, facilitating the removal of the pollutants adsorbed on the modified regenerated cellulose membrane by shearing force.

## 4. Conclusions

Membrane fouling and membrane materials are the main challenges in membrane applications. In this study, an ultrafiltration membrane (ZrO_2_/BCM) was prepared by blending ZrO_2_ particles with natural bamboo cellulose by the phase inversion method. Compared with the original regenerated cellulose membrane, ZrO_2_/BCM has better stability and anti-fouling properties. The water flux of the ZrO_2_/BCM membrane reaches 321.49 (L/m^2^∙h). In the dynamic ultrafiltration experiment, ZrO_2_/BCM showed long-term performance stability, and the membrane flux recovery rate after cleaning with deionized water was 90.6%. In summary, the ZrO_2_/BCM membrane has broader application prospects in the field of water treatment. The membrane prepared in this study can improve the application of degradable membrane material in the green, clean membrane separation process, and promote the sustainable development of communities of human destiny.

## Figures and Tables

**Figure 1 membranes-12-00042-f001:**
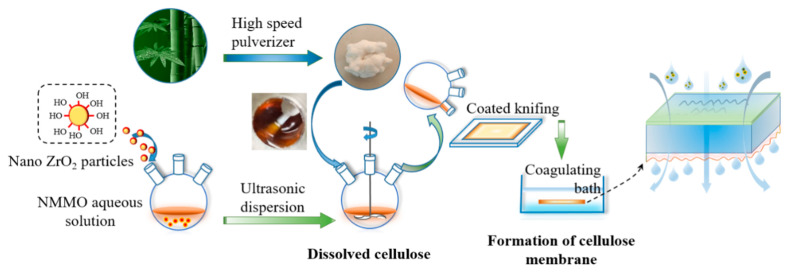
The scheme of the preparation of ZrO_2_/BCM.

**Figure 2 membranes-12-00042-f002:**
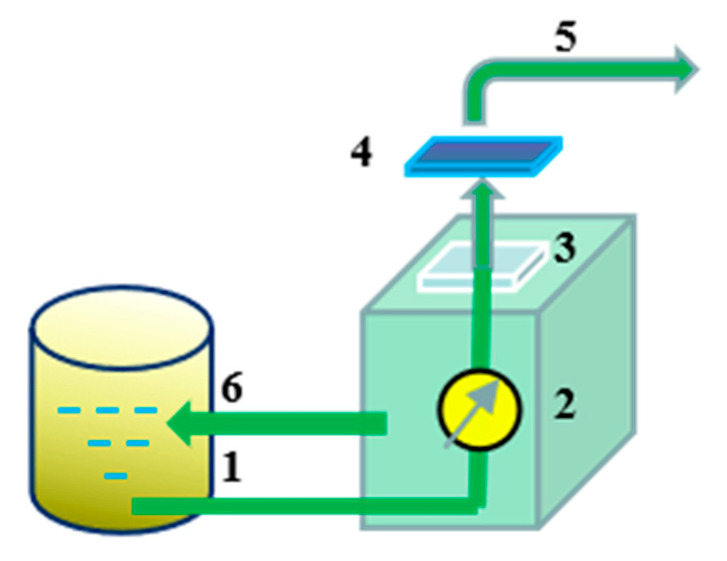
Membrane filtration system: (1) Feed inlet; (2) Pump assembly; (3) Membrane carrier; (4) Membrane; (5) Effluent; (6) Circulating effluent.

**Figure 3 membranes-12-00042-f003:**
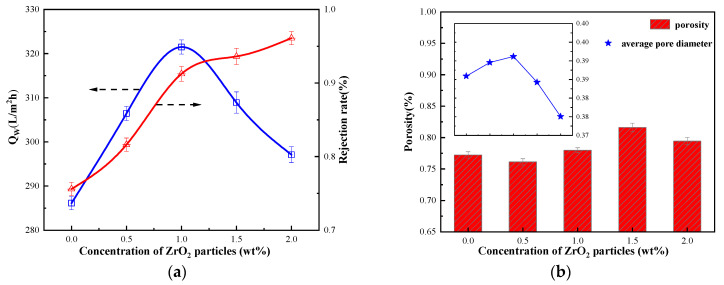
(**a**) Filtration performance of cellulose membrane with nanoparticles; (**b**) Porosity of cellulose membranes with nanoparticles.

**Figure 4 membranes-12-00042-f004:**
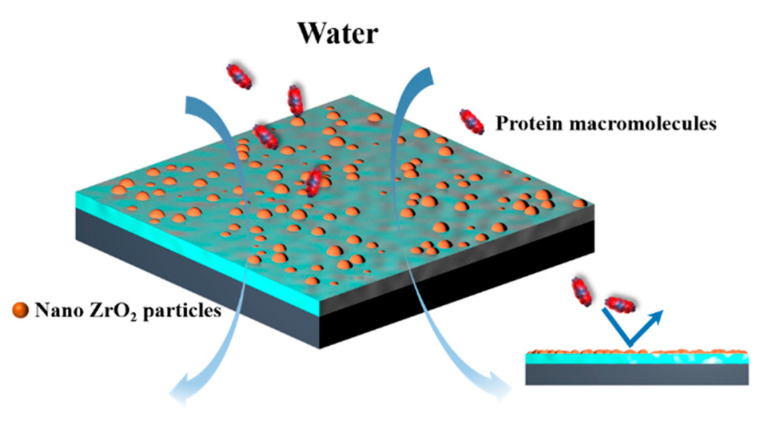
Schematic diagram of ZrO_2_/BCM resistance to contaminant.

**Figure 5 membranes-12-00042-f005:**
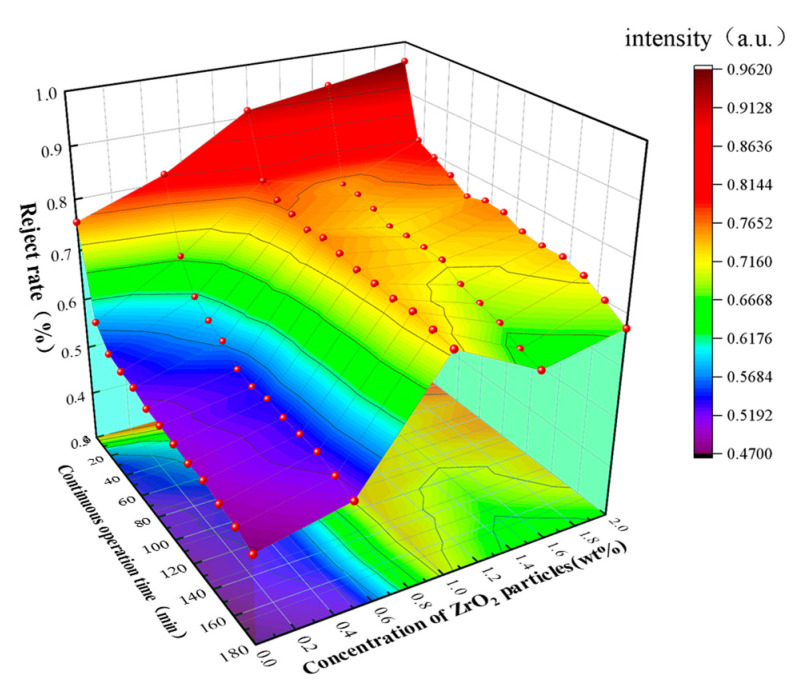
The impact of running time on membrane rejection performance.

**Figure 6 membranes-12-00042-f006:**
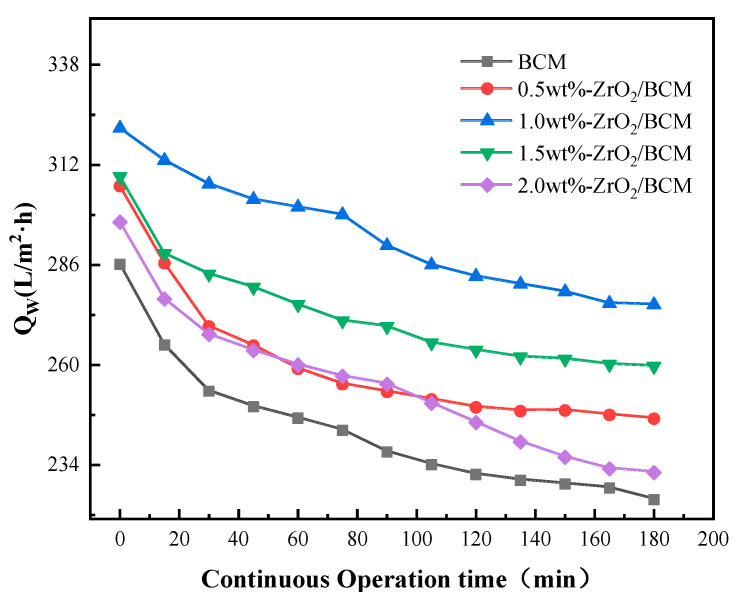
The influence of operation time on the change of membrane flux of BSA solution.

**Figure 7 membranes-12-00042-f007:**
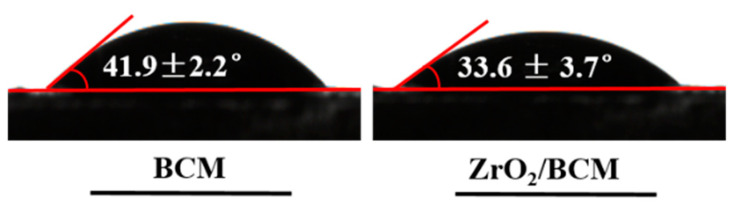
Contact angle images of BCM and ZrO_2_/BCM.

**Figure 8 membranes-12-00042-f008:**
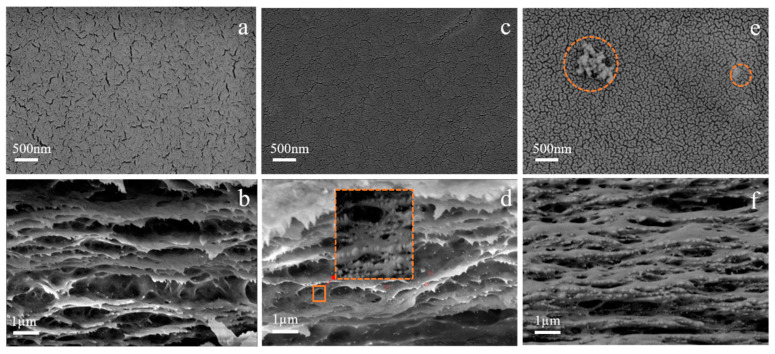
(**a**,**c**,**e**) SEM of the surface of BCM, 1 wt.% ZrO_2_/BCM and 2 wt.% ZrO_2_/BCM; (**b**,**d**,**f**) SEM of the cross section of BCM, 1 wt.% ZrO_2_/BCM and 2 wt.% ZrO_2_/BCM.

**Figure 9 membranes-12-00042-f009:**
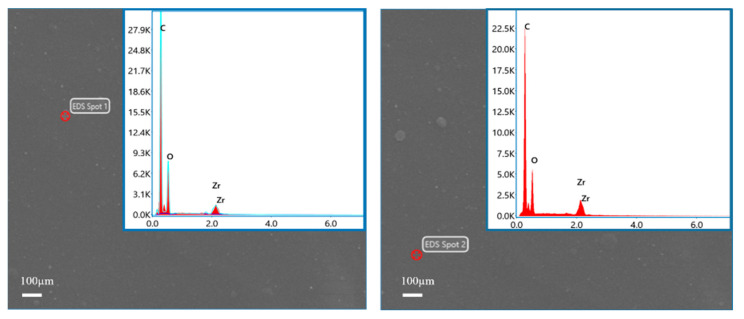
EDs spectrum of 1 wt.% ZrO_2_/BCM.

**Figure 10 membranes-12-00042-f010:**
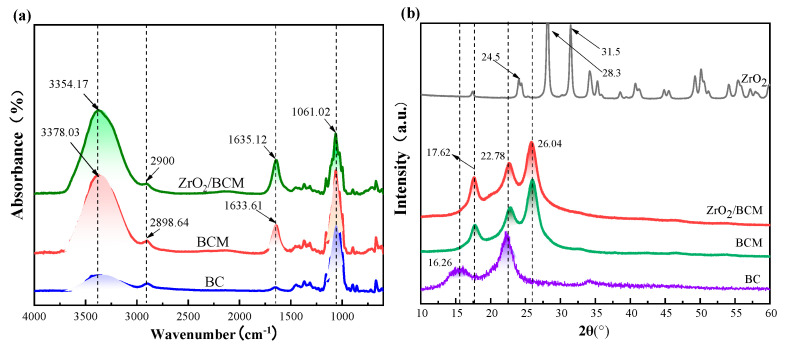
(**a**) FT-IR spectra of regenerated cellulose membrane; (**b**) XRD of BC, BCM, 1 wt.% ZrO_2_/BCM; (**c**,**d**) TG and DTG patterns for BC, BCM, 1 wt.% ZrO_2_/BCM.

**Figure 11 membranes-12-00042-f011:**
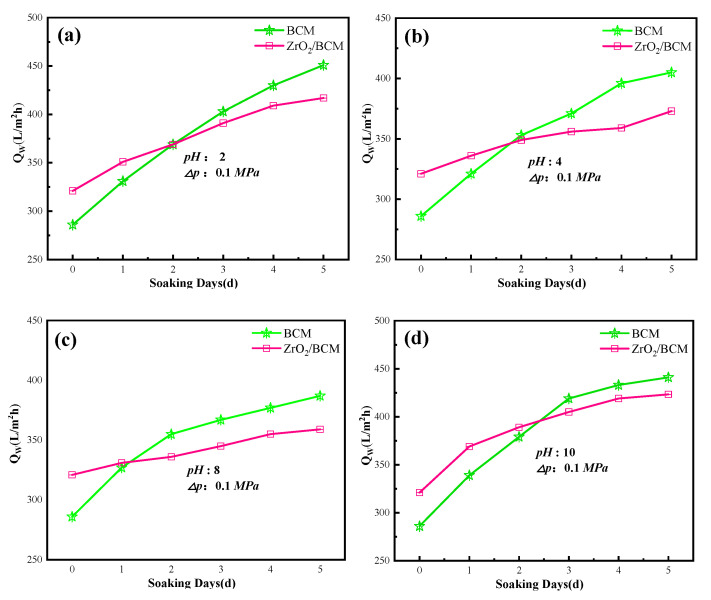
The influence of pH on the membrane fluxes of BCM and 1 wt.% ZrO_2_/BCM. The test conditions (**a**–**d**) of the membrane are respectively: pH 2, 4, 8, 10.

**Figure 12 membranes-12-00042-f012:**
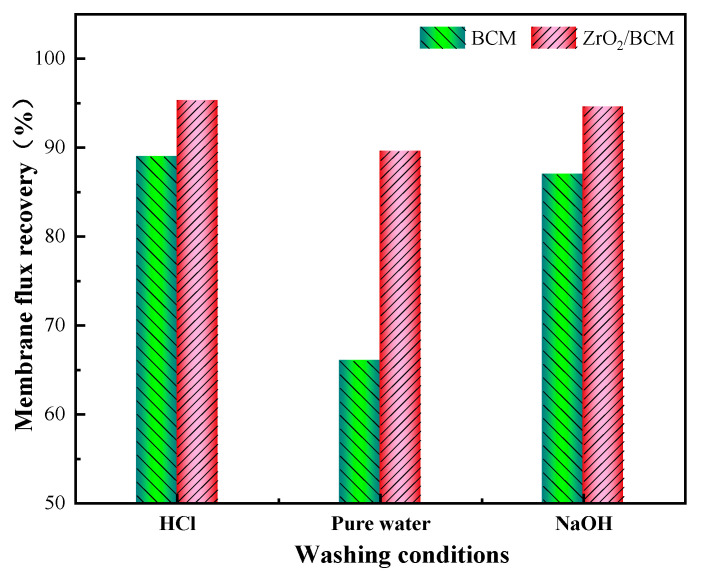
The flux recovery rate of BCM and 1 wt.%-ZrO_2_/BCM.

**Table 1 membranes-12-00042-t001:** Changes of BSA separation efficiency under different ZrO_2_ content.

	BSA Rejection Rate	The Content of ZrO_2_ Particles/wt%				
Operation Time/min		0	0.5	1.0	1.5	2.0
0 15 30 45 60 75 90 105 120 135 150 165 180		0.756 0.578 0.539 0.530 0.526 0.513 0.510 0.505 0.491 0.483 0.472 0.466 0.453	0.816 0.673 0.613 0.589 0.573 0.543 0.536 0.541 0.534 0.533 0.531 0.519 0.507	0.913 0.791 0.771 0.763 0.753 0.759 0.751 0.743 0.741 0.737 0.739 0.732 0.725	0.936 0.761 0.749 0.739 0.724 0.726 0.725 0.723 0.699 0.687 0.675 0.653 0.641	0.961 0.850 0.795 0.776 0.751 0.762 0.759 0.741 0.735 0.736 0.723 0.700 0.673

**Table 2 membranes-12-00042-t002:** Membrane performance parameters of BCM and ZrO_2_/BCM.

Membrane	Water Flux (L/ $m^{2}\cdot h$)	Porosity (%)	Average Pore Diameter (nm)	Contact Angle (°)	BSA Rejection Rate (%)
BCM	286.1 ± 2.7	77.3 ± 2.6	36.5 ± 1.4	41.9 ± 2.2	71.6 ± 2.9
1 wt.%-ZrO_2_/BCM	321.5 ± 4.1	79.8 ± 3.1	39.3 ± 1.5	33.6 ± 3.7	91.2 ± 4.3

## Data Availability

The data presented in this study are available on request from the corresponding author.

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
