# Peer review of "Preparation and Characterization of Regenerated Cellulose Membrane Blended with ZrO2 Nanoparticles"

_membranes, 2021, doi:10.3390/membranes12010042_

Round 1

Reviewer 1 Report

Explanation about the phase conversion process is not given. The pore size of the membrane should be compared to the commercial RO membranes.  

Author Response

We sincerely thank you for completely reviewing our manuscripts and providing very helpful comments. The responses to the comments are given below.

Point 1.“Explanation about the phase conversion process is not given.”

Response: Thanks for the advice. Based on your suggestion, we noticed that the "phase inversion method" (here we use the “inversion” to replace “conversion” for accuracy) proposed in the abstract is not explained in the full text. We first briefly introduce you to the concept of phase inversion.

The "phase inversion method" is referred to as the liquid-solid phase transformation, which is one of the commonly used methods for preparing ultrafiltration membranes. The precursor for the membrane is first dissolved in a solvent and heated to form a homogenous liquid. Afterward, the liquid is uniformly scraped onto the glass plate or non-woven fabric with a scraper, which is further soaked in a non-solvent (mostly pure water) and solidified into a membrane.

To make it clear, we corrected the first sentence at the beginning of "2.2. Preparation of the membrane" in the “Materials and Methods” section into "The cellulose regenerated membrane was prepared by the “liquid-solid” phase inversion method, as schemed in Figure 1." 

Point 2.“The pore size of the membrane should be compared to the commercial RO membranes.”

Response: We discussed and thought about the question you raised. The membranes are usually classified into microfiltration membranes, ultrafiltration membranes, nanofiltration membranes, and reverse osmosis (RO) membranes. There are differences in the pore size and filtration capacity of different types of membranes. The cellulose regeneration membrane prepared in this study belongs to the range of ultrafiltration membranes, and the pore size of the membrane is about 20-40 nm. It can effectively separate proteins, enzymes, and suspend from water. The pore size of the commercial RO membrane is about 0.5-10nm. It can separate heavy metals, residual chlorine, and salt substances. Therefore, the separation performance of the cellulose regenerated membrane prepared in this experiment still has some gaps compared with the commercial RO membrane.

We sincerely hope that this revised manuscript has addressed all your comments and suggestions. We appreciated for reviewers’ warm work earnestly, and hope that the correction will meet with approval. Once again, thank you very much for your comments and suggestions.

Reviewer 2 Report

  1. The formatting should be improved. i.e., line 91-97, there should be a space between number and unit.
  2. More information of the  cellulose membrane should be provided, the thickness and weight were missing in the text.
  3. Fig. 3, it is interesting the tensile strength of the cellulose membrane was increased with the increase of ZrO2 content. Traditionally, the materials mechanical properties should be decrease with the addition of nano-partical, some explanation should be provided.
  4. Did the tensile strength meets the requirements of water treatment? Normally it should provide some standards for comparasion.
  5. Except tensile strength, the tearing strength also should be characterized.

Author Response

We sincerely thank you for completely reviewing our manuscripts and providing very helpful comments. The attachment is our response to your comments, which we would like to submit for your kind consideration.

Reviewer 3 Report

All remarks and comments are highlighted in the attached file with the text of the article. The text, prior to re-evaluation, absolutely requires stylistic and English language correction.

Author Response

We sincerely thank you for completely reviewing our manuscripts and providing very helpful comments. The attachment is our response to your comments, which we would like to submit for your kind consideration.

Thanks for the advice of the reviewer. Thank you for your careful review. We are very sorry for the mistakes in this manuscript and the inconvenience they caused in your reading. We answered questions from your PDF. Please see the attachment for details, the red font is the change.

The responses to the comments are given below. The list of questions is not complete.

  1. Explain “anti-pollution’’

Response: “anti-fouling” instead of “anti-pollution’’. We have deleted the erroneous expression of "anti-pollution" in the full text. "Anti-fouling" is only used to indicate the membrane flux recovery rate after cleaning of polluted membrane and membrane resistance to contaminants.

  1. “What was the thickness of the formed membrane?”

Response: We provide the thickness of the membrane in "2.2. Preparation of the membrane" (see page 3). The revised statement is as follows: Finally, the membrane was taken out and dried in the air to obtain a regenerated cellulose membrane with a thickness of 300 μm.

3.“Eq 1: this is a formula for permeability, not for flux (it takes into account the driving force value)”

Response: Thanks for the advice of the reviewer. We modified the formula of pure water flux in this article according to the marine industry standard of the People's Republic of China "Membrane and module of ultrafiltration" (HY/T112--2008).

4.“what was the range of ZrO2 concentration in the casting solution” and“concentration of BC in the casting solution should be given”

Response: We provide it in "2.2. Preparation of the membrane (line 83)".

5.“What was the BSA concentration? How BSA concentration was measured?”

Response: We provide it in "Performance of Regenerated cellulose Membranes (line 122)".

6.“Fig. 3: What is on the X-axis?”

Response: Fig. 3: X-axis title was added.

  1. “It would be better to show the changes in BSA separation efficiency at different ZrO2 contents in a table or a 2-dimensional graph.”

Response: We have provided a new Table: Table 1. Separation performance of BSA with different ZrO2 content

  1. “Figure 4:no reference to the figure in the text”

Response: Thanks for your suggestion, we have added a reference to Figure 4 in the article.

  1. “These tests(3.4,3.5) were not included in the Materials and methods chapter”

Response: We provided the test method for acid and alkali resistance of regenerated cellulose membrane in "2.4 Performance of Regenerated cellulose membranes", and added the description of the membrane flux recovery rate (r).

Round 2

Reviewer 2 Report

In the author's reply file, the author presented "The tensile strength of the regenerated cellulose film without ZrO2 particles is 0.64 MPa (Figure R1a). The tensile strength of the regenerated cellulose film with 1wt% ZrO2 particles is 0.52 MPa (Figure R1b)."  The data did not match the data of Fig. 3. Please give explanitation.

Author Response

We sincerely thank you for completely reviewing our manuscripts and providing very helpful comments. The responses to the comments are given below.

Point 1. “In the author's reply file, the author presented "The tensile strength of the regenerated cellulose film without ZrO2 particles is 0.64 MPa (Figure R1a). The tensile strength of the regenerated cellulose film with 1wt% ZrO2 particles is 0.52 MPa (Figure R1b)."  The data did not match the data of Fig. 3. Please give  explanation.”

Response: We discussed and thought about the question you raised. The tensile test in Figure 3 in the initial manuscript did not take into account the influence of drying the membrane in the air, which was accounted for by the different experimental data.

After keeping the drying condition consistent, the experimental results in the last reply file showed that adding ZrO2 particles reduced the tensile strength of the cellulose regeneration membrane, as you mentioned. In addition, the tensile strength has no significant influence on the ultrafiltration performance of the membrane in this study. In summary, we deleted the expression about tensile strength in the previous manuscript, and the data in Figure 3(b) only retains the porosity and average pore size.

We sincerely hope that this revised manuscript has addressed all your comments and suggestions. We appreciated for reviewers’ warm work earnestly, and hope that the correction will meet with approval. Once again, thank you very much for your comments and suggestions.

Reviewer 3 Report

The authors use the term "cellulose membrane" and "regenerated cellulose membrane" interchangeably in the text. These are two different materials, and it appears that the research in the paper was conducted for "regenerated cellulose".
It is crucial that the text be linguistically corrected before being reviewed again, as grammatical and stylistic errors often make the text impossible to understand.
All other remarks are marked in the text.

Author Response

We sincerely thank you for completely reviewing our manuscripts and providing very helpful comments. We answered questions from your PDF. Please see the attachment for details, the red font is the change.
The responses to the comments are given below. The list of questions is not complete.

Point 1. “The authors use the term "cellulose membrane" and "regenerated cellulose membrane" interchangeably in the text. These are two different materials, and it appears that the research in the paper was conducted for "regenerated cellulose".
    Response: Thanks for the advice. We have replaced "regenerated cellulose membrane" instead of "cellulose membrane" in the full text. Point 2. “describe the membrane cleaning procedure”

Response: We provide the cleaning procedure of the membrane in "2.4. (4)" (see page 3). The revised statement is as follows: The used regenerated cellulose membrane was placed in the membrane filtration system (Fig 2.) and treated with deionized water, 0.01mol/L HCl, or 0.01mol/L NaOH for 0.5 h, respectively. Then, the membrane was taken out and cleaned with deionized water on both sides several times. The membrane flux recovery rate (r) was obtained by comparing the water flux before and after the cleaning procedure, which was used to evaluate the anti-fouling performance of the regenerated cellulose membrane.

Point 3. Title of Figure 6: “state that the results were obtained with BSA solution”

Response: Thanks for the advice. We have modified the title of Figure 6. The revised statement is as follows: The influence of operation time on the change of membrane flux of BSA solution.

Point 4. Figure 9: “lack of reference in the text”

Response: Thanks for the advice. According to your suggestion, we have added a reference file about the EDs spectrum.

 Point 5. Figure 11: “This information in the figure caption is not needed. In diagrams, specify the unit P

Response: Thanks for the advice. We have modified the diagram.

We sincerely hope that this revised manuscript has addressed all your comments and suggestions. We appreciated for reviewers’ warm work earnestly, and hope that the correction will meet with approval. Once again, thank you very much for your comments and suggestions.

Round 3

Reviewer 3 Report

The corrections made in the text allow for a positive evaluation of the work.